# The Regulatory Network of hnRNPs Underlying Regulating *PKM* Alternative Splicing in Tumor Progression

**DOI:** 10.3390/biom14050566

**Published:** 2024-05-09

**Authors:** Yuchao Li, Shuwei Zhang, Yuexian Li, Junchao Liu, Qian Li, Wenli Zang, Yaping Pan

**Affiliations:** 1Liaoning Provincial Key Laboratory of Oral Diseases, School and Hospital of Stomatology, China Medical University, Shenyang 110002, China; liyuchao@cmu.edu.cn (Y.L.); 20202399@cmu.edu (S.Z.); jcliu@cmu.edu.cn (J.L.); liqian@cmu.edu.cn (Q.L.); 2019120047@stu.cmu.edu.cn (W.Z.); 2Department of Radiation Oncology Gastrointestinal and Urinary and Musculoskeletal Cancer, Liaoning Cancer Hospital & Institute, Cancer Hospital of China Medical University, Shenyang 110042, China; liyuexian@cancerhosp-ln-cmu.com

**Keywords:** hnRNPs, PKM1, PKM2, glycolysis, alternative splicing

## Abstract

One of the hallmarks of cancer is metabolic reprogramming in tumor cells, and aerobic glycolysis is the primary mechanism by which glucose is quickly transformed into lactate. As one of the primary rate-limiting enzymes, pyruvate kinase (PK) M is engaged in the last phase of aerobic glycolysis. Alternative splicing is a crucial mechanism for protein diversity, and it promotes *PKM* precursor mRNA splicing to produce *PKM2* dominance, resulting in low *PKM1* expression. Specific splicing isoforms are produced in various tissues or illness situations, and the post-translational modifications are linked to numerous disorders, including cancers. hnRNPs are one of the main components of the splicing factor families. However, there have been no comprehensive studies on hnRNPs regulating *PKM* alternative splicing. Therefore, this review focuses on the regulatory network of hnRNPs on *PKM* pre-mRNA alternative splicing in tumors and clinical drug research. We elucidate the role of alternative splicing in tumor progression, prognosis, and the potential mechanism of abnormal RNA splicing. We also summarize the drug targets retarding tumorous splicing events, which may be critical to improving the specificity and effectiveness of current therapeutic interventions.

## 1. Introduction

Alternative splicing refers to pre-mRNA exon inclusion or intron exclusion to form alternative splice variants via splicing at different sites. A total of 90% to 95% of human pre-mRNA can produce multiple transcripts by alternative splicing to enrich the diverse proteome. In brief, alternative splicing is the primary way to maintain protein diversity, and preternatural alternative splicing can cause a variety of diseases, including cystic fibrosis, dysautonomia, autoimmune disease, type I diabetes, asthma, and especially cancers [1]. Obviously, compared to normal samples, tumors exhibit up to 30% more alternative splicing events, and thousands of alternative splicing events have only been detected in tumors [2]. Alternative splicing events can control a variety of carcinogenesis processes via cancer hallmarks, including apoptosis, cell cycle, proliferation, metabolism, genomic instability, epithelial–mesenchymal transition (EMT), motility, invasion, angiogenesis, and so on [3]. Alternative splicing is generally divided into the following types: constitutive exon splicing, alternative 5’ splice site, alternative 3’ splice site, cassette exon skipping, retained intron, and mutually exclusive exons, which possess cell and stage-specific characteristics [4]. The process requires splice regulators including serine/arginine-rich proteins (SRs), heterogeneous nuclear ribonucleoproteins (hnRNPs), and RNA binding motif (RBM) families. They belong to RNA binding proteins and recognize different motifs to efficiently affect the splicing decision. SRs have been reported to bind to exon enhancer or intron enhancer sequences more likely to activate splicing, while hnRNPs prefer to bind to exon silencer or intron silencer sequences to suppress the splice site selection. In addition, RBM can also perform similar regulations, but their functions are not entirely clear. Regardless, SRs, hnRNPs, and RBM are essential to tumor oncogenesis and progression.

Oxidative phosphorylation and glycolysis are two main metabolic pathways that provide energy for cells. Normal cells mainly depend on mitochondrial oxidative phosphorylation to generate biological energy. In 1921, Otto Warburg found that cancer cells showed aberrantly active glycolysis even in the presence of sufficient oxygen. Tumor tissues metabolize 10 times the glucose to meet the energy needed for rapid growth, with lactate as the end product. Most cancer cells rely on aerobic glycolysis and the mitochondrial oxidative phosphorylation decreases, which is referred to as the Warburg effect [5,6]. Pyruvate kinase (PK) is involved in the last step of glycolysis and catalyzes phosphoenolpyruvate to pyruvate. There are three isoforms of PK including PKL, PKR, and PKM, among which PKM is the most well-studied. *PKM* precursor mRNA, pre-mRNA, is alternatively spliced into *PKM1* and *PKM2* under multiple splicing factors control. Structurally, *PKM1* excludes exon 10, and *PKM2* excludes exon 9 (Figure 1). PKM1 is primarily expressed in the heart, muscle, brain, and mature sperm, in which cells require a great deal of energy via oxidative phosphorylation. PKM2 is selectively dispersed across proliferating cells with significant anabolic demands, such as embryonic cells, stem cells, gut, and thymus, as well as most tumor cells (Figure 1). The inactive dimer of PKM2 is commonly used by tumor cells [7] to facilitate the transition from oxidative phosphorylation into aerobic glycolysis, which is critical for tumor development [8]. Yet the splicing sites are relatively conservative in *PKM* pre-mRNA, and it cannot fully explain the aberrant splicing events. The upstream regulatory proteins can regulate abnormal expressions of splicing factors or change the interactions with *PKM* pre-mRNA, which eventually drives or affects metabolism programming in cancer cells. Thus, it is crucial to explore the upstream regulatory proteins and relative targets to clarify the aberrant RNA splicing events, which contributes to improving the specificity and effectiveness of cancer therapy. Among the splicing factors regulating *PKM* pre-mRNA splicing, hnRNPs aroused our interest for their multiple regulating approaches. Therefore, we review the structure, mechanism of regulating splicing events, epigenetic effects, and clinical drugs targeting hnRNPA1, hnRNPA2/B1, and hnRNPI/PTBP1.

## 2. SRs, hnRNPs, and RBM Contribute to the Oncogenesis and Tumor Progression via Alternative Splicing

Alternative splicing events are identified as a characteristic of almost all tumors. Abnormal events can generate cancer-related splicing isoforms including *HIF1A, NUMB, PKM, HER2, IRF3, FAS*, *BRCA1*, etc. These isoforms are implicated in boosting cancer cell proliferation, decreasing cell death, enhancing cell migration and metastasis, and renewing metabolism patterns [9]. Splicing factors (SFs) including SRs, hnRNPs, and RBM are able to regulate the splicing events [10,11]. The multiplied mutations and aberrant expressions of SFs are more frequent in cancers. Of note, the SR family mainly contains SRSF1-12, which is usually overexpressed and identified as oncogenes in lung, colon, gastric, ovarian, breast, leukemia, and pancreas tumors [9,12]. SRs are typically bound to exonic splicing enhancers (ESEs) of pre-mRNA and facilitate U1 small nuclear ribonucleic particles (snRNP) and U2 snRNP binding to splicing sites, which contribute to initiating splicing at neighboring splice sites and then including exons. 

The family of heterogeneous ribonucleoproteins (hnRNPs) was first identified in 1965 by Gall et al. [13] and belonged to the most abundant nuclear proteins. hnRNPs are located in the nucleolus of eukaryotic cells and can act as splicing factors facilitating *PKM* pre-mRNAs into *PKM2* mRNAs. There are 21 members in the hnRNPs family [14]. hnRNPs can also promote nucleotide metabolism and serve as oncogenes and suppressors in different types of cancers. In adrenocortical, liver, and lung cancers, hnRNPs show high expressions. However, they show low expressions in kidney cancer and thymoma [15].

The RBM family has been identified as a regulator of alternative splicing factors in the last few decades. The RBM contains 50 members with RNA-recognizing domains in common. Of note, RBM10 is observed as a classic mutation in lung, colon, pancreas, and thyroid cancers [16]. RBM5 is another reported family member with low expressions in lung, breast, and prostate cancers and functions as a tumor suppressor. It has been reported that RBM5 can induce cancer cell apoptosis and inhibit proliferation [17,18]. To function as a splicing regulator, RBM could act on different steps of the splicing events and eventually contribute to cell apoptosis, cell cycle, and other biological process alterations [19]. Up to now, the concrete regulation mechanism has not been fully clarified, and further investigation is required [20,21,22,23].

## 3. The Different Functions Depending on hnRNP Intracellular Localization

The hnRNP family has a molecular weight range of 34–120 kDa and is composed of more than 20 macromolecular proteins as well as a few other small molecule proteins. hnRNPA1, hnRNPA2/B1, hnRNPB2, hnRNPC1, and hnRNPC2 are considered the core members. Except for at least one auxiliary domain that controls protein interactions and subcellular distribution, their protein structures contain one or more RNA binding motifs, which are crucial for hnRNPs acting as RNA-binding proteins. RNP-CS-RBD motif (the most prevalent), RGG box, and KH domain are the three main isoforms of RNA binding motifs. Among these, RNA binding activity in hnRNPA1 is mostly dependent on RNA binding domain (RBD) elements [24]. What’s more, the glycine-rich domain observed in hnRNPA/B is the most prevalent auxiliary domain. Since both hnRNPA1 and A2/B1 have a structure with two RBDs at the N-terminus and one glycine at the C-terminus, they are referred to as 2xRBD-Gly proteins [25]. Due to diverse intracellular localization patterns, hnRNPs exert a variety of functions. On the one hand, RNA splicing, 3′ terminal processing, transcriptional control, and immunoglobulin gene recombination are the principal functions of hnRNPs in the nucleus. On the other hand, mRNA nucleoplasm transport, mRNA localization, metabolism, translation, and protein stability are the main functions of hnRNPs that shuttle between the nucleus and cytoplasm [26]. hnRNPs can also influence how proteins interact with one another and their sub-cellular localization. Apoptosis-related target genes, including *Bcl-2*, *IAP*, and *p53* tumor suppressor genes, as well as exogenous (*Fas*, *caspase-8*, *caspase-2*, and *c-FLIP*) and endogenous (*Apaf-1*, *caspase 9*, and *ICAD*) regulators are negatively impacted by abnormally expressed hnRNPs [27].

## 4. The Regulatory Network of Different hnRNP Family Members

### 4.1. hnRNPA1

hnRNPs were first identified as splicing factors that regulated exon splicing and intron removal to produce a range of transcripts. Additionally, hnRNPs bind to exonic or intronic splicing silencers (ESSs or ISSs) and consequently prevent the binding of splicing factor elements, which diminishes the selection of splicing sites and competes with SRs to promote exon jumping [28]. The hnRNPA/B family members include hnRNPA1, A2/B1, A3, and A0. Among the four members, hnRNPA1 and hnRNPA2/B1 have been studied more widely and deeply compared with hnRNPA3 and hnRNPA0 [29]. Interestingly, hnRNPA2 is structurally like B1, while hnRNPB1 differs from hnRNPA2 in that B1 inserts 12 amino acids at the N-terminal. hnRNPB1 consists of two N-terminus RRMs, a C-terminal LC (containing an RGG cassette), an M9-NLS, and a core PrLD. However, most studies on hnRNPA2/B1 have either failed to distinguish its subtypes or have focused only on A2 [30]. Hereafter, we mainly discussed the structure, distribution, function, and the regulatory network of hnRNPA1 and hnRNPA2/B1 in the hnRNPA/B family.

The hnRNPA1 domain consists of two RNA recognition motifs (RRM1 and RRM2), an RNA binding cassette (RGG), and a nucleus-targeted sequence M9 [31]. The size of the hnRNPA1 protein is about 34 kDa, and its main functions include mRNA splicing, transport, and end particle biosynthesis. hnRNPA1 exhibits minimal expression in normal tissues, but it is highly expressed in various types of malignancies, including breast cancer, prostate cancer, oral cancer, neuroblastoma, bladder cancer, lung cancer, colon cancer, and hepatocellular cancer. As an SF, hnRNPA1 can influence apoptotic gene expressions in cancer cells via regulating alternative splicing, mRNA stability, translation, and protein degradation. In breast and prostate cancer cells, hnRNPA1 weakens programmed cell death by promoting the production of the anti-apoptotic splicing isomer *Bcl-x* and inhibiting the synthesis of the anti-apoptotic protein cIAP1 [32,33]. Additionally, hnRNPA1 can bind to the GUAGUAGU motif found in *CDK2*’s intron 4 region and regulate the inclusion of exon 5 in *CDK2*. In this circumstance, hnRNPA1 controls the expressions of target genes linked to the G2/M stage, further promotes the growth of oral cancer cells [34], and activates telomerase to lengthen telomeres, resulting in the initiation and development of malignant tumors [35]. 

In terms of interfering with glycolysis, the arginine residue of the hnRNPA1 RGG motif binds to the UAGGGC sequence of intron 9 on the side of *PKM* pre-mRNA. hnRNPA1 can facilitate the switch from *PKM1* to *PKM2* in cancer cells, which accelerates glycolysis and cancer initiation [36]. A variety of genes essentially modulate the expression level of hnRNPA1, affecting PKM2 expression and controlling the proliferation, apoptosis, and migration of cancer cells (Figure 2). For instance, MYCN is directly bound to the *hnRNPA1* and *PTBP1* promoter regions and enhances their expressions, respectively. MYCN promotes neuroblastoma cell proliferation and is related to the poor prognosis of patients [37]. Also, hnRNPA1 is regulated in part by certain microRNAs. Let-7a is highly associated with cancer initiation and progression. It has multiple biological functions in cancer cells, including inhibiting cell proliferation, promoting cell differentiation, and apoptosis. However, breast cancer tissues exhibit a low expression of let-7a, preventing *Stat3* translation due to less interaction with the *Stat3* 3′ UTR promoter. Eventually, the low expression of Stat3 reduces the hnRNPA1 level by binding to the GAS in the *hnRNPA1* promoter region. Additionally, let-7a-5p maturation is inhibited by hnRNPA1, thus forming a let-7a-5p/Stat3/hnRNPA1 negative feedback pathway in breast cancer cells [38]. By inhibiting hnRNPA1-dependent *PKM* splicing and consequent *PKM2* overexpression, RBMX neutralizes the aggressive phenotype of metastatic bladder cancer cells. With low expression in metastatic bladder cancer tissues, RBMX can competitively bind to the RGG motif of hnRNPA1 and block it from combining with the lateral intron of *PKM* mRNA exon 9, leading to high expression of *PKM1* and weakening the malignancy and progression of tumors [39]. SAM68, an SRC-related protein in mitosis ubiquitously expressed in lung adenocarcinoma, has been linked to high cancer recurrence frequency, increased cancer-related mortality, and low overall survival. Also, the 351–443 amino acid region of the SAM68 protein is capable of binding to the RGG motif of hnRNPA1, driving *PKM* pre-mRNA alternative splicing onset and increasing *PKM2* expression [40,41]. In addition, other factors also affect hnRNPA1 expression. The HOXB-AS3 peptide can competitively bind to the arginine residue in the hnRNPA1 RGG motif and inhibit the binding of hnRNPA1 to *PKM*, which further down-regulates the level of *PKM2* and inhibits the metabolism reprogramming process of colon cancer cells [42]. A serine protease known as trichosanthes kirilowii protease (TKP) is derived from plants and prevents colon cancer cells from undergoing EMT and promotes apoptosis [43]. It has been reported that TKP significantly reduces β-catenin expression in a dose-dependent manner and inhibits the β-catenin/c-Myc/hnRNPA1 signaling cascade, thereby inhibiting glycolysis and cell proliferation of hepatocellular carcinoma cells [44]. 

### 4.2. hnRNPA2/B1

hnRNPA2/B1 is mainly expressed in breast cancer, colon cancer, prostate cancer, and non-small cell lung cancer, etc. hnRNPA2/B1 functions as alternative splicing and can play a negative role in regulating DNA repair, transcriptional activators, RNA transport, and miRNA expression. hnRNPA2/B1 inhibits 5′ and 3′ splicing site recognition, promotes distal 5′ splicing site selection, and inhibits the use of proximal splicing sites [45]. Several studies have shown that hnRNPA2 binds to the target motif of hnRNPA1 and replaces its splicing role [46]. Exon exclusion and intron inclusion are part of the hnRNPA2/B1 regulating splicing process. Researchers have discovered motifs including UAG(G/A), UAGGG, GGUAGUAG, and AGGAUAGA that could be recognized and implicated in the splicing process [30]. Genes that can be recognized and spliced by hnRNPA2/B1 include *VHLα* [47], *MST1R* [48,49], *c-FLIP*, *BIN1*, *WWOX* [50], *PKM2* [51], etc.

Numerous elements influence hnRNPA2/B1 expression or its function to increase the production of PKM2, improving the glycolytic capacity of cancer cells. Uncoupling proteins (UCP), which are extensively expressed in a variety of cancer tissues, are anionic transporters at the inner mitochondrial membrane (IMM). By boosting the anti-apoptotic characteristics, UCP can inhibit the accumulation of mitochondrial ROS, which leads to chemotherapy resistance and enhances tumor aggressiveness. However, the specific concentration of ROS triggering this tumor-promoting effect is not yet determined. Through its antioxidant role, UCP2 increases the expression of hnRNPA2/B1 and upregulates GLUT1 and PKM2, increasing glycolytic activity and lactate production in breast cancer cells (Figure 3) [51]. Several non-coding RNAs can control hnRNPA2/B1 expression. The *hnRNPA2/B1* mRNA 3′UTR region’s translation stability is maintained by miR-369, and its overexpression shifts from *PKM* to *PKM2* variable to facilitate metabolic reprogramming [52]. In contrast, miR-124 and miR-137 accelerate the conversion of *PKM* pre-mRNA to *PKM1* in colon cancer cells by inhibiting hnRNPA2. In this scenario, glucose is primarily metabolized through oxidative phosphorylation to inhibit colon cancer cell growth [53]. 

Exosomes carrying lncRNA LNMAT2 are secreted by prostate cancer cells, and these exosomes interact with hnRNPA2/B1 to facilitate lymph angiogenesis and lymph node metastases [54]. In non-small cell lung cancer, c-Myc can bind to the LINC01234 promoter to increase its transcription. LINC01234 then interacts with hnRNPA2/B1 to create a new carcinogenic loop of c-Myc/LINC01234/hnRNPA2/B1/miR-106b-5p/CRY2/c-Myc [55]. LncRNA H19 is upregulated in colon cancer and associated with a poor prognosis. H19 directly binds to hnRNPA2/B1 and afterward initiates the ERK signaling pathway, which orchestrates an EMT in colon cancer cells [56]. Also, a variety of lncRNA HOTAIR sequence segments can bind hnRNPA2/B1, especially the B1 subtype, which has a strong affinity for HOTAIR. Additionally, hnRNPB1 binds to chromatin and preferentially correlates with HOTAIR transcripts, promoting the invasion capability of breast cancers [57].

### 4.3. PTBP1

hnRNPI, also known as polypyrimidine tract-binding protein 1 (PTBP1), interacts with the polypyrimidine present at the upstream branch point of exons to serve as an SF in most situations. It consists of 531 amino acids, 4 RNA-binding RRM domains, and nuclear shuttle domains at the N terminus [58]. PTBP1 has a molecular weight of 59 kDa, which makes it easy to attach to promoters. It is highly expressed in tumor tissues and linked to a poor prognosis of colon cancer [59], glioma [60], renal clear cell carcinoma [61], and anaplastic large cell lymphoma [62]. PTBP1 increases alternative splicing events of *CD44* to produce *CD44* v8-10, which promotes cancer cell invasion. In addition, PTBP1 can also boost the expressions of cell-cycle-related proteins such as cyclin A, cyclin B, cyclin D, cyclin E, and CDC2, to promote tumor cell proliferation [63]. To accelerate tumor growth, PTBP1 develops radiation and chemotherapy resistance and has a positive correlation with hypoxic lesions [64].

The main roles of PTBP1 are splicing localization and poly-acylation of mRNA. PTBP1 can serve as an SF for many genes, including *MEIS2*, *PKM* [65], *Axl* [66] and *EXOC7*, etc. [67]. The binding of PTBP1 to pre-mRNA inhibits the splicing of exons adjacent to the binding sites and improves binding to the optimal motif in the polypyrimidine beam near the 3′-terminal splicing site (e.g., UCUUC). This action prevents the inclusion of downstream exons as a consequence [68]. PTBP1 preferentially binds to intron 8 when it comes to *PKM* pre-mRNA, further excluding exon 9, which leads to exon 10 inclusion and ultimately promotes *PKM2* transcription (Figure 4) [61]. The expression of PTBP1 is markedly increased by EGF, which also promotes tumorigenesis by increasing transcription and translation of *PKM2* [69]. MTR4 is a nuclear exosome-associated RNA helicase, and functions as an essential component of RNA processing and surveillance. It is an independent diagnostic marker for hepatocellular carcinoma patients with poor prognosis. In the promoter region of *MTR4*, c-Myc binds to 2 non-classical E cassettes (CACGCG, CACGAG) and 695 bp CpG islands around the TSS, recruiting PTBP1 to the target pre-mRNA through an independent RNA and protein interaction. This guarantees proper alternative splicing processes and produces the glycolytic gene *PKM2* to boost cancer metabolism [70].

In addition, several tissue-specific non-coding RNAs can act as upstream targets of PTBP1. miR-124, miR-1, miR-133b, miR-137, miR-206, and miR-340 inhibit PTBP1 expression and reduce the PKM2/PKM1 ratio, impeding glycolysis [53,71] (Table 1). The brain contains a high level of miR-124, which has been extensively investigated in neurodevelopment. miR-124 has anti-tumor effects by increasing the oxidative stress products activated by the tricarboxylic acid cycle (TCA) and causing apoptosis and autophagy [72]. Particularly in the brain and colorectal cancers, low expression of miR-124 triggers a feedback cascade on PTBP1/PKM1/PKM2, which promotes cancer cell growth [73]. PKM2 could be significantly reduced by si-PTBP1 or miR-124 mimics, demonstrating that these molecules were crucial in controlling the ratio of PKM2/PKM1 [74]. miR-1 and miR-133 also function as anti-tumor non-coding RNAs expressed in muscle tissues, which mediate cell autophagy by silencing PTBP1. In addition, miR-133b can directly downregulate the pathogenic gene PAX3-FOXO1, causing reduced PTBP1 expression. All of this results in a dominant manner of PKM1-modulated oxidative phosphorylation in cells [75]. Other tissue-specific microRNAs can also regulate PTBP1 expression and stabilize the downstream target genes, which may participate in controlling the *PKM* splicing networks. For instance, PTB-AS stabilizes the mRNA by binding to the 3′UTR region of *PTBP1* and increasing *PTBP1* expression [76]. In non-small cell lung cancer, miR-644a functions as a tumor suppressor to inhibit PTBP1 expression [77], and in colorectal cancer, HuR binds to the 3’UTR region of *PTBP1* to promote *PTBP1* stability. However, circRHOBTB3 promotes HuR ubiquitination degradation caused by β-Trcp1 and reduces the production of downstream PTBP1 [78].

## 5. Epigenetic Modifications of hnRNPs Affect Glycolysis of Tumor Cells by Regulating PKM2

Under certain circumstances, continuous accumulations of genetic changes and epigenetic modifications in key tumor suppressor genes and oncogenes affect the occurrence and development of tumors [79]. Epigenetic modification refers to changes in gene expression rather than altering DNA sequence [80]. Common epigenetic changes in tumors include abnormal DNA methylation, histone modifications, and altered expression levels of various non-coding RNAs. PKM2 can be mutually regulated through phosphorylation, acetylation, and other modifications, which change PKM2 intracellular localization and specific biological functions, including energy supply for cancer cells, EMT, cell proliferation, invasion, and metastasis (Table 2) [81]. However, attention must also be paid to the epigenetic alterations of the upstream SFs causing the shift in PKM2 expression. The stability or affinity of hnRNPs is affected by modifications like acetylation, phosphorylation, and ubiquitination. Ultimately, it influences whether cells use PKM1-based oxidative phosphorylation metabolism or PKM2-based glycolysis.

The acetylated and phosphorylated hnRNPA1 increases its affinity with *PKM* pre-mRNA promoting *PKM2* isoform generation and enhances cancer cell proliferation and invasion. In lung adenocarcinoma, ESCO2, as an evolutionarily conserved cohesion acetyltransferase, can catalyze hnRNPA1 acetylated at K277 and retain hnRNPA1 in the nucleus. This increases hnRNPA1 binding to *PKM* EI9 together with more *PKM2* isoform mRNA production [82]. S6K2 catalyzes the phosphorylation of hnRNPA1 in Ser6 and causes colorectal cancer cells to preferentially express PKM2 instead of PKM1 and enhance the cell proliferation activity [83]. However, de-acetylation and ubiquitination increase PKM1 isoforms and inhibit aggressive cancer cell isotypes. In hepatocellular carcinoma, SIRT1 and SIRT6 cause de-acetylation at K3, K52, K87, and K350 lysine residues in hnRNPA1, hindering the splicing transition from *PKM* to *PKM2* mRNA. This weakens PKM metabolic activity and the non-metabolic PKM2/β-catenin signaling pathway, which in turn inhibits cancer cells from utilizing glycolysis and proliferating [84]. E3 ubiquitin ligase ZFP91, a tumor suppressor, is typically less expressed in hepatocellular carcinoma. Overexpression of ZFP91 can promote K48 polyubiquitination of hnRNPA1 at K8, leading to hnRNPA1 degradation via the proteasome pathway [85]. And for PTBP1, its de-acetylation can decrease its affinity for *PKM* pre-mRNA. SMAR1, PTBP1, and HDAC6 can form a ternary complex, and in this scenario, SMAR1 can catalyze PTBP1 to maintain the deacetylated state in an HDAC6-dependent manner. SMAR1 inhibits glucose uptake and lactate production by reducing PKM2 expression, therefore inhibiting breast cell metabolism and malignancy [86].

In addition to glucose metabolism, the post-translation modifications of hnRNPs exert other aspects on cellular metabolism. The ubiquitination of PTBP1 and hnRNPA2/B1 affect lipogenesis in intrahepatic cholangiocarcinoma [87] and hepatocellular carcinoma [88]. Additionally, hnRNPK phosphorylation influences mRNA metabolism [89]. So far, other types of metabolism are not as extensively investigated as glucose metabolism regulated by the hnRNPs family. More research on other metabolic effects needs to be explored in the future.

## 6. Potential Therapeutic Drugs or Inhibitors

Many effective drugs targeting PKM2 have already been found, such as benserazide (BEN), benzoxepane derivatives 10i, ML-265, parthenolide, TT-232 (CAP-232), TEPP-46, combohydroquinone, and histone deacetylase inhibitors (HDACi). These inhibitors can reduce mitochondrial metabolic reprogramming, weaken cell proliferation, inhibit tumor cell growth, and mediate apoptosis [74,90,91,92,93]. In addition to these existing inhibitors, exploration of medicine targeting the SFs of PKM to decrease PKM2 production may have great prospects to impair the glycolysis in cancer cells.

As one of the essential SFs for PKM, PTBP1 is highly expressed in two drug-resistant colon cancer cell lines (HCT-8/V and HCT116). Knocking down PTBP1 enhances the chemotherapy sensitivity and leads to inhibition of glycolysis [94]. Several plant-derived ingredients have been reported to show anti-tumor benefits. Kaempferol is essentially a flavonoid in fruits and vegetables that induces apoptosis and inhibits colon cancer cell proliferation in a dose-dependent manner. In addition, kaempferol promotes miR-339-5p expression, which directly targets hnRNPA1 and PTBP1, reducing PKM2 expression and inhibiting glycolysis in colon cancer [95]. Moreover, another plant-derived oleanolic acid induces a transition from PKM2 to PKM1 by inhibiting mTOR signaling and the c-Myc-dependent expressions of hnRNPA1 and hnRNPA2, thus weakening the glycolytic ability of cancer cells [96]. 

hnRNPs are correspondingly downregulated after the application of inhibitors. Quercetin is a flavonoid abundant in plants that specifically binds to the C-terminal region of hnRNPA1. Quercetin impairs the ability of hnRNPA1 to shuttle between the nucleus and cytoplasm and ultimately traps it in the cytoplasm [33]. β-caprylylone is the main component of coriander volatile oil, a Chinese herbal medicine that has recently been shown to have anti-glioma effects. Previous studies have shown that β-caprylylone can inhibit the expression of hnRNPA2/B1. β-caprylylone thereby promotes *Bcl-x* alternative splicing, increasing the ratio of Bcl-xS/Bc and mediating apoptosis of glioma cells [97]. In addition, pretreatment with the synthetic drug cilostazol, which is a novel antiplatelet drug, can reduce the overexpression of hnRNPA2/B1 in human dermal microvascular endothelial cells [98]. Nanoparticle-coupled aptamers bind specifically to hnRNPA2/B1, identifying and inhibiting the proliferation of a variety of tumor cells (HepG2, MCF-7, H1299, and HeLa), and they may have a promising application in cancer diagnosis and treatment [99]. The above findings suggest that hnRNP inhibitors are expected to become antineoplastic agents, and combining with PKM2 agonists or inhibitors may achieve better efficacy (Table 3). However, all the drugs or inhibitors mentioned above are only investigated in in vitro cells or animal models, the clinic trial information is limited and not reported yet. The effective therapeutic drugs are expected to be studied for clinical use in the future.

## 7. Conclusions and Future Perspectives

It has been decades since *PKM* alternative splicing first became implicated in tumorigenesis and progression. With the development of new methods for analyzing the alternative splicing events on a big scale, including quantAS [100], isoform long-reads RNA-seq (Iso-seq), and single-cell RNA-seq [101], our understanding of this event will continue to grow. Despite the emphasis on the change from PKM1 to PKM2, it does not seem that it can entirely account for the variety of splicing events. New knowledge may also provide novel targets for hnRNPs like hnRNPA1, hnRNPA2/B1, PTBP1, etc. We currently provide a brief review of the most recent research on the upstream regulators of hnRNPs, as well as the developed target drugs and inhibitors in vitro. In the future, more studies will be conducted to explore the unknown information on hnRNPs members. Additionally, there is scope for more investigation into deeper mechanisms of aberrant alternative splicing events in malignancies, as well as the potential side-effects of the aforementioned medicines and inhibitors. Small molecules, splice-switching antisense oligonucleotides, CRISPR-based approaches, or engineered small nuclear RNAs, etc. may be used for modulating the splicing events.

## Figures and Tables

**Figure 1 biomolecules-14-00566-f001:**
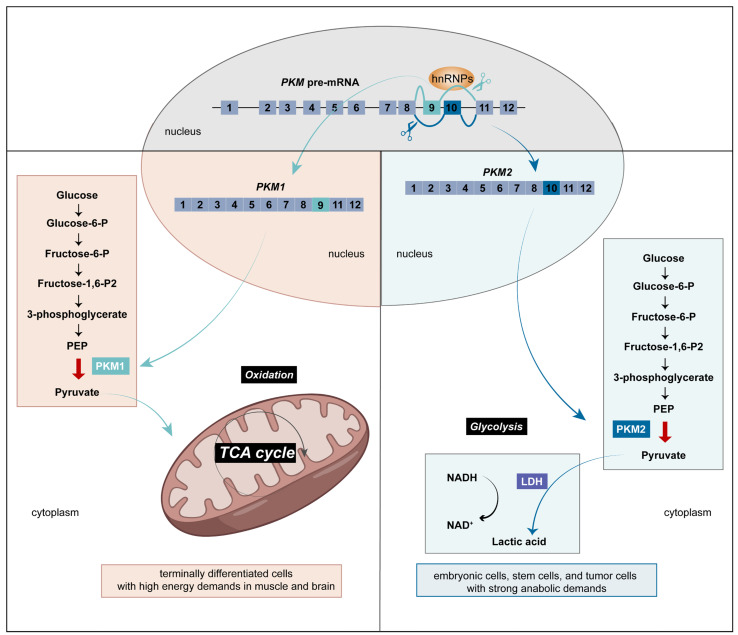
Schematic representation of different *PKM* pre-mRNA splicing patterns. (**Left**): In muscle and brain, terminally differentiated cells mainly depend on oxidative phosphorylation to generate biological energy in mitochondria. During this process, splicing factors exclude exon 10 of *PKM* pre-mRNA to generate more *PKM1*. (**Right**): In rapidly proliferating cells, such as embryonic cells, stem cells, and tumor cells, splicing factors exclude exon 9 and contain exon 10 to generate more *PKM2* for strong anabolic demands.

**Figure 2 biomolecules-14-00566-f002:**
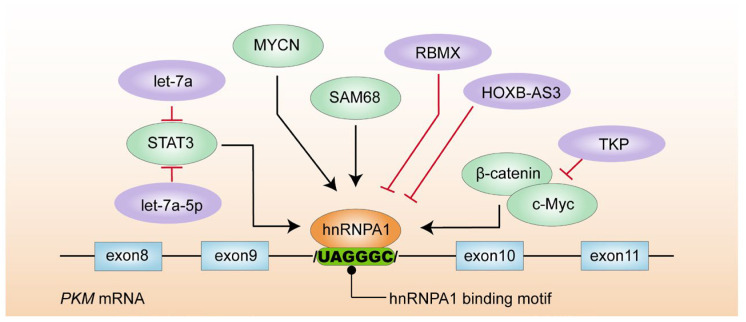
The graph depicts a regulatory network of hnRNPA1 in *PKM* alternative splicing in cancer cells.

**Figure 3 biomolecules-14-00566-f003:**
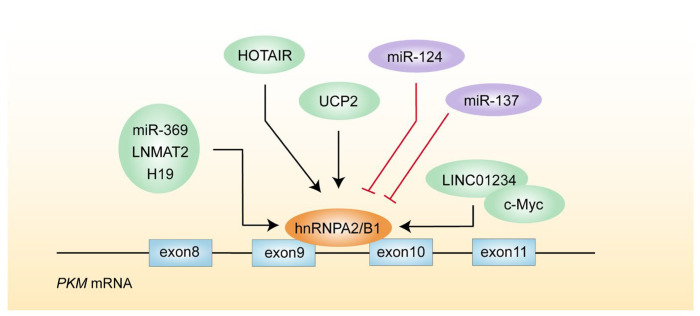
The graph depicts a regulatory network of hnRNPA2/B1 in *PKM* alternative splicing in cancer cells.

**Figure 4 biomolecules-14-00566-f004:**
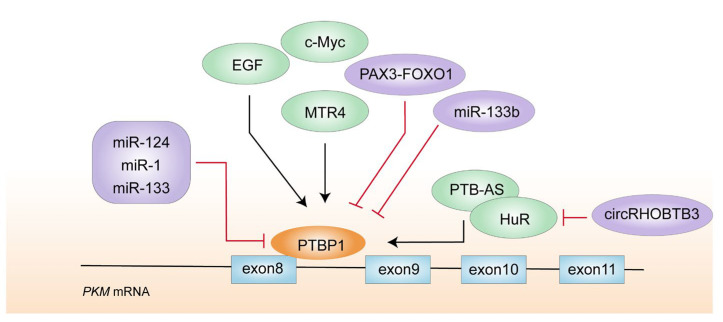
The graph depicts a regulatory network of PTBP1 in *PKM* alternative splicing in cancer cells.

**Table 1 biomolecules-14-00566-t001:** Non-coding RNA regulating PTBP1 in *PKM* alternative splicing events.

Non-Coding RNA Partner	Mechanism	Function/Effect	Type of Cancer	Refs
miR-124	inhibits PTBP1 expression and reduces PKM2/PKM1 ratio	triggers a feedback cascade about PTBP1/PKM1/PKM2 and has anti-tumor effects	brain and colorectal cancers	[72,73,74]
miR-1 miR-133	silences PTBP1	leads to a high level of *PKM1* mRNA	rhabdomyosarcoma	[75]
miR-133b	downregulates the pathogenic gene PAX3-FOXO1	reduces PTBP1 expression	rhabdomyosarcoma	[75]
PTB-AS	binds to the 3′UTR region of PTBP1	stabilizes and increases PTBP1 expression	glioma	[76]
circGLIS3	sponge miR-644a and PTBP1 can bind to the flanking introns of cirGLIS3	circGLIS3/miR-644a/PTBP1 positive feedback loop and promotes *PTBP1* production	non-small cell lung cancer	[77]
HuR	binds to the 3’UTR region of *PTBP1*	promotes *PTBP1* stability	colorectal cancer	[78]
circRHOBTB3	promotes HuR ubiquitination degradation	reduces the production of downstream PTBP1	colorectal cancer	[78]

**Table 2 biomolecules-14-00566-t002:** The epigenetic regulation of hnRNPs in *PKM* alternative splicing events.

	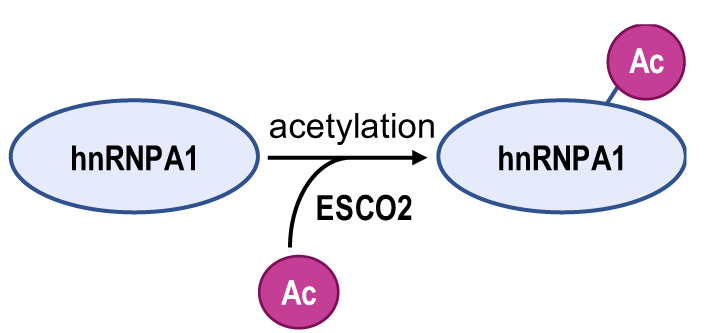	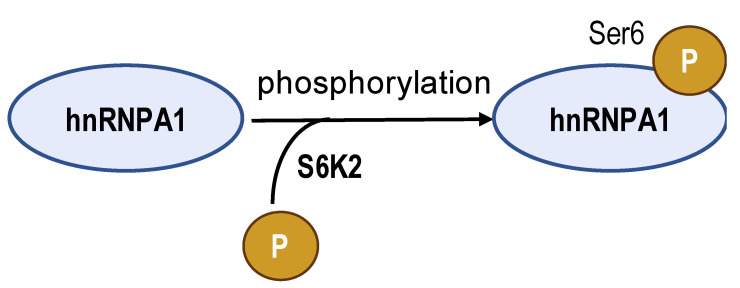	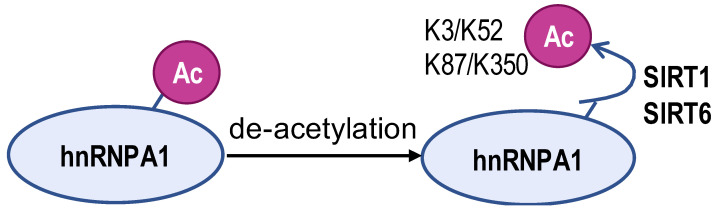	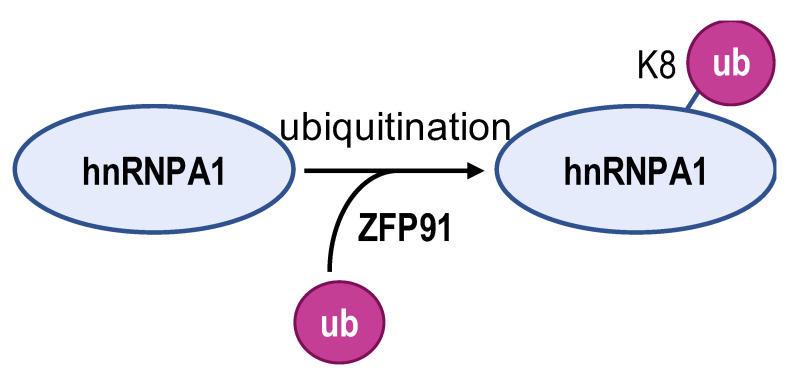	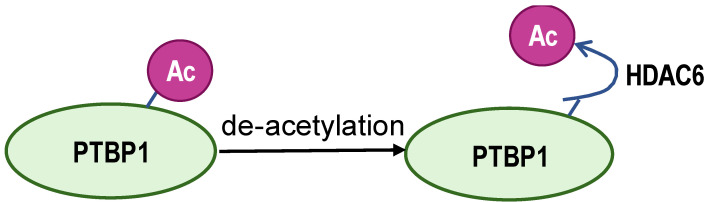
**Enzyme**	ESCO2	S6K2	SIRT1 and SIRT6	ZFP91	HDAC6
**Modification sites**	K277	Ser6	K3, K52, K87, and K350	K8	unreported
**Mechanism**	retains hnRNPA1 in the nucleus and increases hnRNPA1 binding to *PKM* EI9	promotes hnRNPA1 binding to the splicing site of the *PKM* gene	weakens PKM metabolic activity and PKM2-β-catenin pathway	hnRNPA1 is degraded by the proteasome pathway	SMAR1, PTBP1, and HDAC6 can form a ternary complex
**PKM2/PKM1 ratio**	upregulated	upregulated	downregulated	downregulated	downregulated
**Type of cancer**	lung adenocarcinoma	colorectal cancer	hepatocellular carcinoma	hepatocellular carcinoma	breast cancer
**Refs**	[82]	[83]	[84]	[85]	[86]
**Outcome**	**Pro-tumor**		**Anti-tumor**		

**Table 3 biomolecules-14-00566-t003:** Drug or inhibitor agents applied in regulating hnRNP expression.

Drugs or Inhibitors	Therapeutic Target	Molecular Mechanisms	Cancer Characteristics	Refs
kaempferol	miR-339-5p	promotes miR-339-5p expression which downregulates hnRNPA1 and PTBP1	colon cancer	[95]
oleanolic acid	mTOR signaling	inhibits the c-Myc-dependent expression of hnRNPA1 and hnRNPA2	prostate carcinoma and breast cancer	[96]
quercetin	hnRNPA1	impairs the ability of hnRNPA1 shuttling between the nucleus and cytoplasm and ultimately traps it in the cytoplasm	prostate cancer	[33]
β-caprylylone	hnRNPA2/B1	inhibits the expression of hnRNPA2/B1	glioma	[97]
cilostazol	hnRNPA2/B1	reduces the overexpression of hnRNPA2/B1	Bechet’s disease	[98]
nanoparticle-coupled aptamer aptamers	hnRNPA2/B1	acts as a cancer-specific probe	hepatocellular carcinoma; breast cancer; non-small cell lung cancer; cervical cancer	[99]

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
