# Peer review of "The Regulatory Network of hnRNPs Underlying Regulating PKM Alternative Splicing in Tumor Progression"

_biomolecules, 2024, doi:10.3390/biom14050566_

Round 1
Reviewer 1 Report
Comments and Suggestions for Authors
This is a well-structured review on the regulatory role of heterogenous ribonucleopreoteins (hnRNPs) over pyruvate kinase (PK) M. the approach is novel, and the subject is relevant for a field where epigenomics and metabolism merge. The contents are appropriate and well documented, with bibliography relatively up to date (2023). As a nice touch, it briefly points out other pathways affected by these PTM (i.e lipogenesis). In section 6, I would add any clinical trials if avaliable. The schemes are detailed and adequate to represent the text. The tables are good in content but in my opinion, could use a style polish, just for clarity. In the conclusions I miss a brief reference to the state-of-the-art to detect these changes and how it will affect future works.
There are some minor comments/typos:
223 intrachondria: I rather use inner mitochondrial membrane (IMM)
226 When it says, "accumulate mitochondrial ROS, which leads to chemotherapy resistance and enhances tumor aggressiveness", is arguable: I would specify that there is a window where generates resistance. We have to consider that there is a threshold, after which the accumulation of ROS is toxic ad is used as mechanism of action for several drugs (i.e. melatonin).
310 Even though there are some heritable epigenetic modifications, this is not common to all of them. The most relevant fact of these modifications is how they translate environmental cues in changes of gene expression. In my opinion, "heritable" here, drives confusion in the context of this sentence. I rather remove heritable from the sentence. Also, the full sentence needs a citation.
375 "defend": Review the word choice
Comments on the Quality of English Language
I would have it review for a native speaker.
Reviewer 2 Report
Comments and Suggestions for Authors
Line 39: The abbreviation "EMT" should be replaced with the full name as this is the first place being mentioned.
Line 51: This paragraph misses a topic/leading sentence. For example, you can start with "Oxidative phosphorylation and glycolysis are two metabolic pathways that provide energy for cells."
Line 60: "PKM1 contains exon 9 and PKM2 contains exon 10." This is confusing. Sounds like they only contain one exon or contain 9-10 exons. Consider replacing it with "PKM1 excludes the exon 10, and PKM2 excludes the exon 9."
Line 103-106: Need citations.
Line 203: "The hnRNPA/B family members include hnRNPA1, A2/B1, A3, and A0". Move this sentence to the gap between lines 145 and 146, and add more description. In other words, having a brief paragraph to generally introduce HnRNP Family Members before discussing each of them in different subsections.
For Fig 1, 3, and 4, start to refer to those figures immediately when possible. For example, in lines 60-61, Fig 1 should be referred for the first time: "Structurally, PKM1 contains exon 9 and 60 PKM2 contains exon 10 (Fig 1)."
